# INDOOR 3.6M: A BENCHMARK IMAGE DATASET FOR GLOBAL INDOOR GEOLOCATION

## ABSTRACT

Image geolocation has achieved strong progress for outdoor imagery, powered by large-scale datasets and distinctive visual cues such as landmarks, skylines, and natural environments. Indoor geolocation, however, remains largely unexplored: indoor scenes lack persistent geographic structure, exhibit limited contextual range, and are severely underrepresented in existing benchmarks. To address these challenges, we introduce INDOOR-3.6M, the first large-scale dataset dedicated to indoor geolocation, featuring diverse indoor scenes collected from multiple global sources together with enriched metadata and multimodal annotations. We further propose a geographically representative sampling framework that jointly incorporates population, land area, and visual diversity to mitigate geographic bias. Using this dataset, we train state-of-the-art geolocation models—including PIGEON, GeoCLIP, and Translocator—and evaluate them across coarse and fine-grained spatial scales using accuracy and distance-based metrics. Our results show that coarse indoor geolocation is feasible, with models achieving strong continent-level accuracy, while fine-grained indoor localization (city and street level) remains highly challenging. We release INDOOR-3.6M along with an indoor image benchmark test set (INDOOR-40K). Along with our evaluations, these establish a foundation for future research in indoor geolocation.

## 1 INTRODUCTION

Image geolocation—determining a photograph's geographic origin from visual content (Hays & Efros, 2008)—is critical for applications such as forensic investigations and fraud detection. Existing approaches follow either retrieval-based methods, which match query images against geotagged databases (Hays & Efros, 2008; Vo et al., 2017), or classification-based methods, which discretize Earth's surface into geocells and treat geolocation as a multi-class prediction task (Seo et al., 2018; Weyand et al., 2016). More recently, hybrid systems that integrate both paradigms have emerged, exemplified by state-of-the-art methods such as PIGEON (Haas et al., 2023) and GeoCLIP (Vivanco Cepeda et al., 2024), which leverage CLIP Vision Transformers (Dosovitskiy et al., 2020; Radford et al., 2021) trained on large-scale geotagged datasets.

As in other vision domains—including object detection, segmentation, and scene recognition—progress in geolocation hinges on large, diverse, and high-quality datasets. ImageNet (Krizhevsky et al., 2017), MS COCO (Lin et al., 2014), and Places (Zhou et al., 2017) have each provided transformative advances. For geolocation, this need is even more pronounced: visual

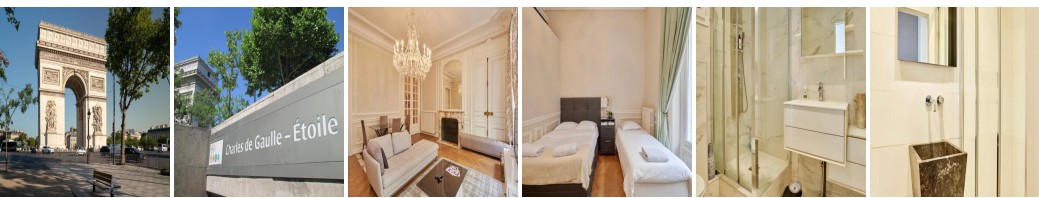

Figure 1: Images (a) and (b) show outdoor landmarks near the Arc de Triomphe, while (c)–(f) depict diverse indoor scenes from a nearby hotel. This highlights the geolocation challenge posed by visually similar indoor environments compared to distinctive outdoor environments.

appearance varies with season, weather, time of day, human modification, and architectural style (Pramanick et al., 2022). Achieving global coverage further requires capturing heterogeneous visual characteristics across regions, while fine-grained localization demands dense geotagged imagery to support precise spatial inference.

Despite rapid progress in outdoor and mixed-environment geolocation, indoor geolocation remains substantially under-explored. Outdoor scenes benefit from distinctive cues—landmarks, signage, skylines, and natural features—whereas indoor environments are visually constrained, often repetitive, and offer limited contextual range. Buildings, rooms, and enclosed spaces typically lack the broad spatial context available outdoors, and variations in design, layout, and lighting introduce additional challenges. These factors underscore the need for indoor-specific datasets.

Indoor images depict enclosed or semi-enclosed spaces—such as homes, offices, commercial buildings, public facilities, and transitional areas like parking garages or covered patios—that are shaped by architectural style, functional purpose, and cultural context. Although these environments lack the expansive contextual cues available outdoors, they contain geographically informative signals: regional differences in materials, furniture and decor, cultural and religious artifacts, electrical fixtures, interior design conventions, and spatial organization all reflect local norms and socioeconomic factors. These stylistic and structural variations provide subtle but meaningful geographic cues, making coarse-scale indoor geolocation feasible despite the constrained and often repetitive nature of indoor scenes.

Given the field's heavy emphasis on outdoor imagery and the paucity of indoor-focused datasets, a geographically diverse and representative indoor geolocation dataset is essential. Such a resource would capture the variability of indoor environments across the world and provide the foundation needed to study indoor-specific challenges. To advance research in this direction, we make the following contributions:

- We introduce **INDOOR-3.6M**, a large-scale dataset of 3.6 million geotagged diverse indoor images with global coverage and enriched metadata, designed specifically to support research on indoor geolocation.
- We propose a **geographically representative sampling framework** that combines visual diversity, land area, and population statistics.
- We release **INDOOR-40K**, a geographically representative benchmark test set for indoor geolocation.
- We conduct the first **comprehensive indoor geolocation benchmark** across hybrid, retrieval, and vision–language reasoning paradigms. We train GeoCLIP, PIGEON, and Translocator-inspired architectures on INDOOR-3.6M and evaluate their performance at continent ($\leq 2500km$), country ($\leq 750km$), region ($\leq 200km$), city ($\leq 25km$), and street level ($\leq 1km$).

To the best of our knowledge, our evaluations provide the first systematic characterization of indoor geolocation difficulty. Indoor-specific training yields substantial gains at coarse scales, yet fine-grained indoor localization remains unresolved across all model families, with city- and street-level accuracy remaining extremely low. The dataset and evaluation scripts are available at: `https://github.com/anonymous-for-double-blind-review`.

## 2 RELATED WORK

Image geolocation has advanced rapidly through cutting-edge computer vision techniques, deep learning architectures, and large-scale geotagged datasets. The field has been shaped by two primary paradigms: retrieval-based approaches that match query images with similar images in geotagged databases (Hays & Efros, 2008; Vo et al., 2017), and classification-based approaches that divide Earth's surface into discrete geocells (Weyand et al., 2016), treating geolocation as multi-class classification. Recently, hybrid approaches (Astruc et al., 2024) have emerged, which integrate retrieval and classification methods to overcome the limitations of geographic discretization. This integration is typically achieved using sophisticated training techniques such as ranking losses, contrastive learning objectives, or a two-stage approach involving classification followed by regression.

State-of-the-art systems like PIGEON/PIGEOTTO (Haas et al., 2023) and GeoCLIP (Vivanco Cepeda et al., 2024) exemplify these advances. These models utilize CLIP Vision Trans-

Table 1: Comparison of geolocation datasets. The "Benchmark" column indicates whether the dataset provides a dedicated test set specifically designed to evaluate geolocation model performance.

| Dataset | Year | Size | Scene Type | Scale | Type | Benchmark |
|---|---|---|---|---|---|---|
| Im2GPS | 2008 | 6.5M | Mixed | Global | – | – |
| YFCC100M | 2016 | 100M | Mixed | Global | Multimodal | – |
| MP-16 | 2017 | 5M | Mixed | Global | – | – |
| PlaNet | 2016 | 126M | Outdoor | Global | – | – |
| Hotels50K | 2019 | 1M | Indoor (Hotels) | Global | – | – |
| OpenStreetView-5M | 2024 | 5.1M | Outdoor | Global | – | – |
| Im2GPS3k | 2017 | 3K | Mixed | Global | – | ✓ |
| YFCC4k | 2017 | 4K | Mixed | Global | – | ✓ |
| YFCC26k | 2018 | 26K | Mixed | Global | – | ✓ |
| GWS15K | 2023 | 15K | Outdoor | Global | – | ✓ |
| **INDOOR-3.6M** | **2024** | **3.6M** | **Indoor (Diverse)** | **Global** | **Multimodal** | – |
| **INDOOR-40K** | **2024** | **40K** | **Indoor (Diverse)** | **Global** | **Multimodal** | ✓ |

formers (ViTs) (Dosovitskiy et al., 2020; Radford et al., 2021) and leverage large-scale geotagged datasets to infer geographic locations from visual content. Their success highlights the effectiveness of modern neural architectures in capturing complex visual features tied to specific locations, and the importance of combining such architectures with comprehensive, high-quality datasets.

Recent work has further expanded geolocation capabilities across multiple dimensions. GOMAA-Geo (Sarkar et al., 2024) introduces goal-modality agnostic active geo-localization, where an agent uses sequential aerial observations to localize targets specified via aerial, ground-level, or textual descriptions. OpenStreetView-5M (Astruc et al., 2024) presents a large-scale, globally distributed street-view dataset of over five million localizable images that advances outdoor geolocation benchmarking. Around the World in 80 Timesteps (Dufour et al., 2024) proposes a generative approach to global visual geolocation based on diffusion and flow-matching methods, modeling full probability distributions over locations. GaGA (Dou et al., 2024) formulates an interactive geolocation paradigm and develops a multimodal large-language-model assistant that iteratively refines predictions using dialogue and the MG-Geo dataset. G3 (Jia et al., 2024) introduces a retrieval-augmented framework for worldwide geolocalization, combining multi-modal alignment of images, GPS coordinates, and textual descriptions with diversified prompting and verification. These advances demonstrate the field's rapid evolution toward more flexible, interactive, and capable geolocation systems.

Existing benchmark datasets such as IM2GPS (Hays & Efros, 2008) and IM2GPS3k (Vo et al., 2017), along with YFCC100M subsets like YFCC4K (Vo et al., 2017) and YFCC26K (Muller-Budack et al., 2018), have been instrumental in evaluating geolocation systems (Table 1). However, these datasets predominantly comprise outdoor imagery, rendering them inadequate for indoor-specific geolocation assessment. Indoor environments present unique challenges, necessitating interpretation of more complex and nuanced visual features including room layout variations, furniture arrangements, lighting conditions, and decorative elements. To facilitate accurate indoor geolocation, specialized indoor-specific datasets for both training and benchmarking are essential.

Indoor datasets like NYU Depth V2 (Silberman et al., 2012), SUN RGB-D (Song et al., 2015), and Places365 (Zhou et al., 2017) are designed for object detection and scene recognition but lack the geographic metadata necessary for geolocation. Similarly, mixed-environment datasets such as MediaEval Placing Task (MP-16) (Larson et al., 2017) and YFCC100M (Thomee et al., 2016), which encompass both indoor and outdoor environments, also fall short for indoor geolocation. While Hotels-50K (Stylianou et al., 2019) provides indoor imagery, it is limited to hotel rooms and lacks the scene diversity needed for general indoor geolocation. While Hotels-50K Stylianou et al. (2019) provides indoor imagery, it is restricted almost entirely to hotel rooms and therefore lacks the scene diversity required for general indoor geolocation.

INDOOR-3.6M addresses this critical gap by providing a large-scale, scene-agnostic indoor geolocation dataset, enabling models to learn fine-grained features across diverse indoor environments worldwide.

## 3 DATASET OVERVIEW

INDOOR-3.6M provides **scene-agnostic** coverage spanning diverse residential, commercial, public, industrial, and transitional spaces worldwide, rather than being limited to predefined scene types or specific categories. This diversity enables geolocation models to generalize across environments and learn fine-grained features critical for accurate prediction.

**Data Sources and Collection:** INDOOR-3.6M was assembled from three primary sources – **Flickr** (flickr.com, 2024), **Wikidata** (wikidata.org, 2024), and **Booking.com** (booking.com, 2024), spanning 223 countries worldwide. We queried images using indoor scene categories from Places365[1], restricted to Creative Commons licenses and GPS coordinates. To improve retrieval, we expanded search terms to include broader indoor-related keywords such as "indoor", "interior", "lobby", and "room". Crucially, these terms served only to aid retrieval, not as filtering constraints. All candidates were filtered using the Places365 indoor/outdoor classifier, retaining only images with $P(\text{indoor}) \geq 0.5$. Figure 2 illustrates the visual and semantic diversity of INDOOR-3.6M. Samples are grouped by indoor-likelihood score $P(\text{indoor})$ (rows) and by Level-2 Places365 indoor scene categories (columns), showing the progression from borderline to strongly indoor environments across a wide range of scene types. Country and source labels further highlight the dataset's global and cross-platform coverage.

**Filtering and Deduplication:** Visual inclusion was thus determined solely by indoor likelihood rather than predefined scene membership. Finally, we remove redundancy using (i) pHash-based duplicate filtering, (ii) spatial–temporal clustering (lat/lon rounded to $10^{-3}$, month–year, and Places365 scene type) with one image retained per cluster. These steps are applied prior to splitting, ensuring that the benchmark test set does not contain visual-duplicate or room-level overlaps with the training data.

**Scale and Distribution:** While the dataset aims to represent diverse indoor environments, it is not entirely geographically uniform due to reliance on inherent biased online sources (Figure 3a). Regions with strong digital footprints and larger populations (e.g., United States, representing 30% of the data) are over-represented, while areas with less online activity or smaller populations are under-represented.

**Metadata Enrichment:** Each image was reverse-geocoded using the Nominatim API (Nominatim, 2024) to obtain address information. Accompanying textual metadata—including user tags, descriptions, and captions—was collected when available. To support multimodal research, we provide additional visual annotations: (1) top-10 scene category labels from Places365 and a ViT model trained on MIT Indoor Scenes; (2) segmentation masks from the Segment Anything Model (Kirillov et al., 2023); and (3) object detection results from YOLOv8 (Jocher et al., 2023). These automated annotations are not manually curated and may contain noise. The annotations facilitate identification of geolocation-relevant features such as products, furniture, and cultural artifacts—consistent with real-world applications like Europol's "Trace an Object" initiative[2], where objects within images aid location inference. While we preserve original images unaltered in our evaluations, we release SAM segmentation masks and YOLOv8 detections to enable downstream applications to perform selective masking (e.g., faces or persons) for privacy-preserving workflows without modifying the raw images. Automated scene and object labels are not manually curated and may contain noise.

## 4 INDOOR IMAGE GEOLOCATION BENCHMARK DATASET

We introduce **INDOOR-40K**, a spatially representative benchmark test set specifically designed for indoor geolocation evaluation. It is sampled directly from INDOOR-3.6M images captured after 2017 to ensure temporal separation from the widely-used pretraining datasets like YFCC100M yielding an initial pool of approximately 800,000 candidate images from all three sources (Flickr, Wikidata, and Booking.com). We apply the sampling methodology described next to construct a geographically representative subset of 40,000 images across 232 countries and 6 continents.

---

[1]`https://github.com/CSAILVision/places365/blob/master/categories_`
`places365.txt`
[2]`https://www.europol.europa.eu/stopchildabuse`

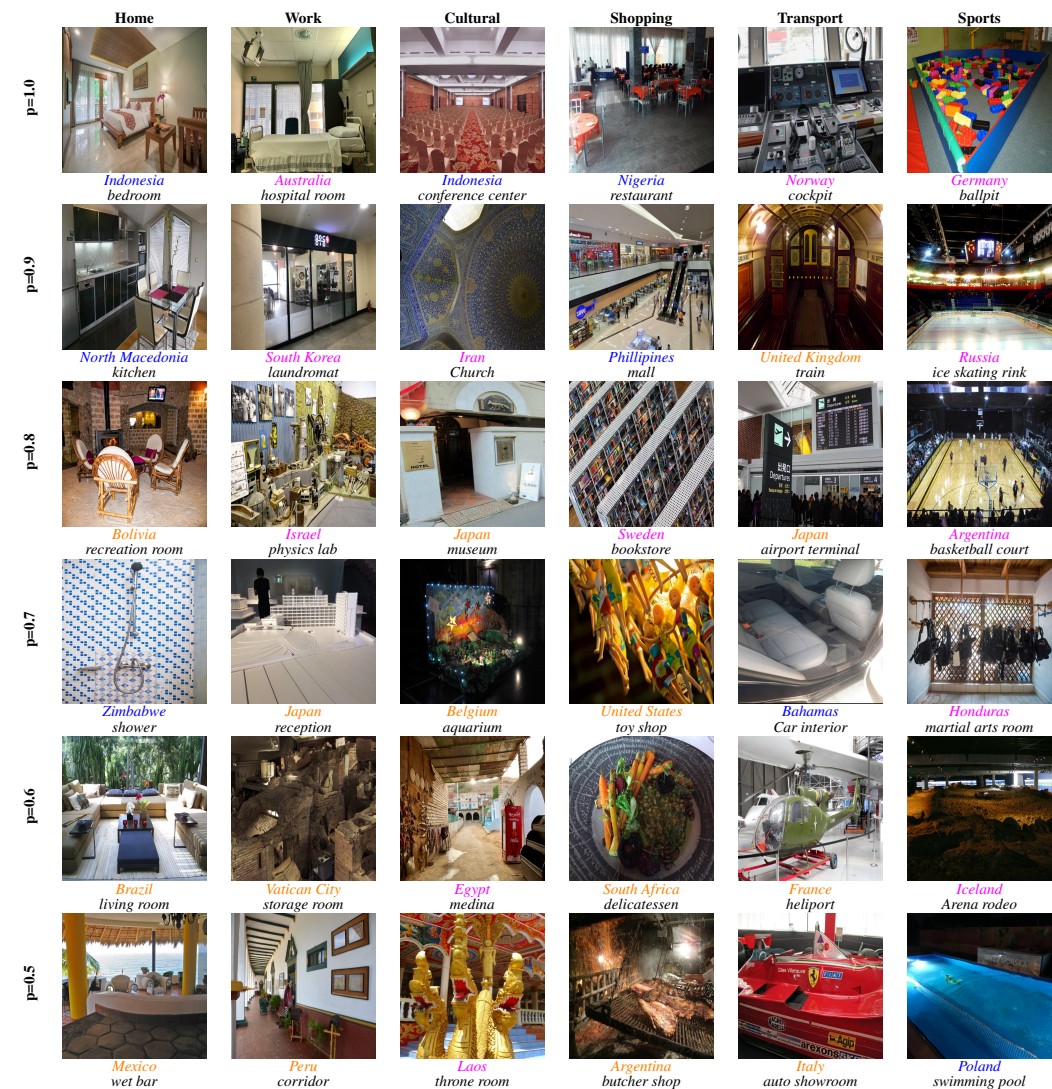

Figure 2: Samples of images from the dataset representing different parts of the world. The rows correspond to the indoor likelihood score $P(\text{indoor})$, while the columns categorize the scene types. Country names in blue, magenta, and orange are sourced from Booking.com, Wikidata, and Flickr, respectively. Scene labels and indoor likelihood scores are provided using Places365 classifier.

## 4.1 SAMPLING STRATEGY

Our sampling strategy incorporates geographic and visual factors into subset construction. The method proceeds in three steps. First, we compute visual diversity scores from image embeddings obtained using the CLIP ViT-L/14. Diversity is quantified by the average pairwise cosine distance between embeddings, stratified across semantic scene categories within each country. Second, we incorporate external country-level statistics—land area and population size—into a regression framework that estimates their relative contributions to observed visual diversity. The resulting weights are expressed as: $w_c = \alpha \cdot \text{Population}_c + \beta \cdot \text{LandArea}_c$, where $w_c$ is the sampling weight for country $c$. Several regression models were evaluated; Random Forest regression achieved the highest performance ($R^2 = 0.861$) in predicting country-level diversity, substantially outperforming linear and polynomial models. Feature importance analysis further indicated that both factors contributed nearly equally (population: $0.493$, land area: $0.507$).

Finally, samples are drawn proportionally to country weights $w_c$ across scene categories, yielding a geographically representative dataset (Figure 4). We applied this sampling strategy to both our

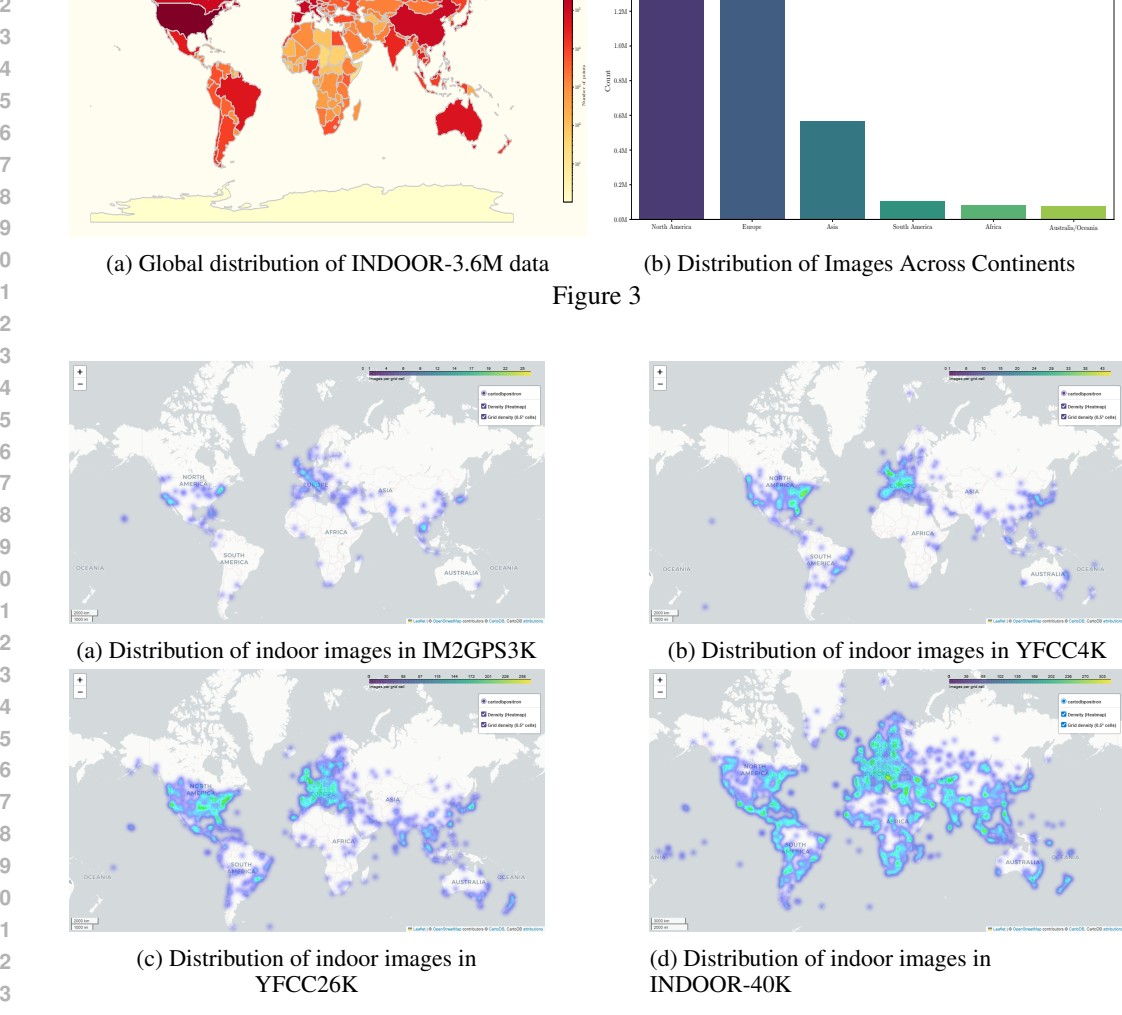

(a) Global distribution of INDOOR-3.6M data    (b) Distribution of Images Across Continents

Figure 3

(a) Distribution of indoor images in IM2GPS3K    (b) Distribution of indoor images in YFCC4K

(c) Distribution of indoor images in YFCC26K    (d) Distribution of indoor images in INDOOR-40K

Figure 4: Indoor image distributions ($p_{indoor} \geq 0.5$) across geolocation benchmarks. Existing datasets (IM2GPS3K, YFCC4K, YFCC26K) are geographically skewed, while Indoor40K (ours) achieves more representative global coverage.

benchmark test set and training subsets from INDOOR-3.6M, ensuring geographic and scene-level diversity throughout the evaluation pipeline. We perform ablation studies comparing model performance on datasets sampled with and without our method.

## 5 INDOOR GEOLOCATION BENCHMARK

This section establishes the first unified benchmark for global indoor geolocation. Indoor scenes lack many of the global cues that make outdoor geolocation tractable, raising a fundamental question: *is indoor visual geolocation even feasible from visual features alone?* We begin by assessing this feasibility using a *retrieval-based* approach using pretrained visual encoders and nearest neighbour retrieval. These models provide a method for measuring how much geographic signal is present in indoor imagery before any indoor-specific supervision is introduced.

Building on this, we then evaluate established modern visual geolocation paradigms to understand how well current methods transfer to the indoor setting.

Our evaluation reveals three key findings: **(1) Continental-scale indoor geolocation is feasible**, with models trained on INDOOR-3.6M achieving 66% continent accuracy. **(2) Strong pretrained visual encoders provide competitive zero-shot baselines**, with CLIP-style models reaching 50–59% continent accuracy without any geolocation-specific training. **(3) Fine-grained localization remains an open challenge**, with street-level accuracy reaching only 3.6% for GeoCLIP* and 0.4% for both PIGEON* and Translocator*.

## 5.1 EXPERIMENTAL SETUP

We evaluate three methodological paradigms introduced above:

- **Retrieval Models:** Similar to Astruc et al. (2024), frozen visual encoders (CLIP, DINOv2, ResNet, StreetCLIP, IndoorCLIP) are evaluated via nearest-neighbour search on INDOOR-40K. (Table 2).
- **Hybrid Models:** GeoCLIP*, PIGEON*, and Translocator* trained on INDOOR-3.6M, using the training modifications described below. Table 3 summarizes the performance on INDOOR-40K.
- **Vision–Language Models:** LLaVA-1.6 and InternVL2 evaluated in a zero-shot setting via constrained text prompts for predicting continent and country labels. Table 4 reports zero-shot performance of these models as well as a random baseline.

We additionally evaluate cross-dataset generalization of the best performing models to indoor subsets from im2gps3k, yfcc4k, and yfcc26k. (Table 5).

**Metrics.** Following geolocation practice, we report geolocation accuracy as the percentage of test images whose predicted GPS coordinates fall within the corresponding distance thresholds at five spatial scales: Continent ($\leq$ 2500 km), Country ($\leq$ 750 km), Region ($\leq$ 200 km), City ($\leq$ 25 km), and Street ($\leq$ 1 km), as well as mean and median geodesic error for continuous predictions.

**Training Details.** The retrieval models used remain frozen throughout. GeoCLIP*, PIGEON* and Translocator* are trained from scratch on images from indoor 3.6M. For GeoCLIP*, we replace the GeoCLIP image encoder with IndoorCLIP. Additionally, we use separate learning rates for the two encoders: the image encoder is updated with ($\texttt{img\_lr} = 1 \times 10^{-3}$) and follows a cosine annealing schedule, while the location encoder uses ($\texttt{loc\_lr} = 1 \times 10^{-5}$). We also retain the original GPS gallery introduced in (Vivanco Cepeda et al., 2024) for sampling negative location pairs. For Translocator*, we follow the dual-stream design of the original Translocator architecture, which processes each RGB image together with its corresponding segmented image in parallel encoders. We replace the HRNet-based segmentation masks used in the original paper with masks generated by SAM. In addition, the original formulation includes an auxiliary Places365 indoor/outdoor(urban)/urban(landscape) classifier to provide additional environmental signals. Since INDOOR-3.6M contains only indoor imagery, these targets lose their semantic meaning; we therefore replace this auxiliary head with a Places365 scene-classification objective.

Table 2: Zero-shot retrieval performance of visual encoders using nearest-neighbor search on INDOOR-40K.

| # | Architecture | Size (M) | Pretraining | | Mean(km)↓ | Median(km)↓ | Distance Accuracy (%)↑ | | | | |
|---|---|---|---|---|---|---|---|---|---|---|---|
| | | | Objective | Dataset | | | Continent | Country | Region | City | Street |
| 1 | ResNet-50 | 23 | Classification | ImageNet-1k | 6611.9 | 6325.1 | 28.5 | 11.2 | 6.9 | 5.6 | 4.7 |
| 2 | ResNet-50 | 23 | Classification | Places365 | 6696.0 | 6473.4 | 27.5 | 10.4 | 6.1 | 5.0 | 4.3 |
| 3 | ViT-B/32 | 88 | CLIP | LAION-2B | 5563.0 | 4127.4 | 40.5 | 20.7 | 13.7 | 11.4 | 9.7 |
| 4 | ViT-L/14 | 300 | DINOv2 | DINOv2 | 5953.0 | 5047.2 | 35.9 | 16.8 | 10.4 | 8.3 | 6.8 |
| 5 | ViT-L/14 | 300 | CLIP | LAION-2B | 4651.7 | 2472.4 | 50.1 | 28.4 | 19.4 | 15.7 | 13.3 |
| 6 | ViT-L/14 | 300 | CLIP | DataComp | 4292.7 | 2062.2 | 54.0 | 31.8 | 21.6 | 17.3 | 14.4 |
| 7 | ViT-L/14 | 300 | CLIP | MetaCLIP | 4242.3 | 2037.9 | 54.4 | 31.8 | 21.3 | 17.05 | 14.16 |
| 8 | ViT-L/14 | 300 | CLIP | OpenAI | 4349.7 | 2144.6 | 53.2 | 30.4 | 20.1 | 15.9 | 13.1 |
| 9 | ViT-L/14 | 300 | StreetCLIP | OpenAI+GSV | 3969.9 | 1797.2 | 57.1 | 34.5 | 23.4 | 18.7 | 15.4 |
| 10 | ViT-L/14 | 300 | IndoorCLIP (ours) | Indoor3.6M | **3843.9** | **1665.6** | **59.0** | **36.6** | **25.2** | **20.2** | **17.1** |

Table 3: Performance of classification and hybrid methods on INDOOR-40K. GeoCLIP* and PI-GEON* are trained on INDOOR-3.6M. Higher accuracy and lower mean distance are better.

| Model | Continent (%) | Country (%) | Region (%) | City (%) | Street (%) | Mean (km) |
|---|---|---|---|---|---|---|
| Translocator* | 25.1 | 7.7 | 2.6 | 1.2 | 0.4 | 6928.2 |
| PIGEON* | 63.9 | 33.9 | 12.2 | 2.2 | 0.4 | **3037.1** |
| GeoCLIP (pretrained) | 51.8 | 24.0 | 10.0 | 4.8 | 2.0 | 4316.2 |
| GeoCLIP* | **66.1** | **40.3** | **22.3** | **11.5** | **3.8** | 3178.2 |

Table 4: Zero-shot VLM performance on INDOOR-40K. Models predict continent and country via text generation without retrieval or fine-tuning.

| Model | Continent (%) | Country (%) |
|---|---|---|
| LLaVA-v1.6-Vicuna-7B | 35.2 | 3.2 |
| InternVL2-8B | **50.1** | **5.5** |
| Random Guess | 16.67 | 0.43 |

## 6 DISCUSSION

**Retrieval models reveal strong pretrained geographic priors.** Frozen visual encoders provide a direct measure of how much geographic signal is present in indoor imagery. CLIP-style models achieve 50–59% continent accuracy (Table 2), with StreetCLIP reaching 57.1% and the lowest zero-shot median error (1,797 km). IndoorCLIP improves upon StreetCLIP across all spatial scales, reducing mean and median error by 126 km and 132 km, respectively, and increasing accuracy by 1.5–2.1 percentage points. These results show that indoor-specific text–image alignment yields measurable benefits.

**Hybrid models achieve the strongest overall performance.** GeoCLIP* is the best-performing model across most scales, reaching 66.1% continent accuracy and 40.5% country accuracy (Table 3). Relative to pretrained GeoCLIP (51.8% continent), indoor-specific training gives a substantial improvement. PIGEON* performs competitively at coarse scales (63.9% continent) but degrades sharply at finer ones, with 0.4% street-level accuracy. Translocator* performs poorly across all scales. These outcomes indicate that alignment-based hybrid models transfer more effectively to indoor data than classification- or segmentation-driven designs. TransLocator* performs substantially worse than other hybrid baselines on our indoor benchmark, which is consistent with its architectural design. The model's dual-stream formulation integrates a semantic segmentation branch that was introduced to stabilize outdoor geolocation under large appearance changes such as lighting, weather, and seasonal variation. In indoor settings, where appearance is comparatively stable across seasons, this segmentation branch contributes limited discriminative signal and may introduce noise when applied to scenes outside its training domain. In contrast, alignment-based models such as GeoCLIP* do not rely on segmentation priors and remain more robust indoors.

**Fine-grained indoor localization remains difficult.** Across retrieval and hybrid approaches, performance drops steeply with granularity. Even the best-performing model, GeoCLIP*, reaches only 11.0% accuracy at the city level and 3.6% at the street level, while PIGEON* and Translocator* achieve 0.4% at street scale. This highlights the present limitations of existing geolocation architectures for precise indoor prediction.

**Vision–language models offer coarse but limited predictions.** InternVL2-8B achieves 50.1% continent accuracy (Table 4) but only 5.5% country accuracy, indicating that current VLMs can identify broad regional patterns but lack the spatial precision of retrieval and hybrid models.

**Indoor-specific supervision generalizes beyond INDOOR-40K.** GeoCLIP* outperforms outdoor-pretrained GeoCLIP across indoor subsets of existing benchmarks, reaching 66.7–71.3% continent accuracy on im2gps3k, yfcc4k, and yfcc26k (Table 5). These results show that training on INDOOR-3.6M yields improvements that persist beyond the curated INDOOR-40K test set.

Taken together, our findings establish that coarse indoor geolocation is feasible with current methods, while fine-grained indoor localization remains a challenging open problem.

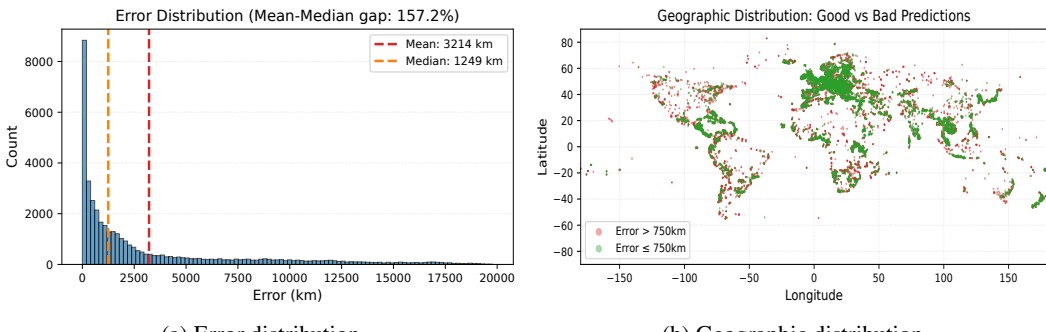

(a) Error distribution      (b) Geographic distribution

Figure 5: Regional and scene-based performance analysis.

Table 5: Performance of models trained on INDOOR-3.6M evaluated on indoor subsets from existing mixed-environment benchmarks.

| Dataset | Model | Cont. | Country | Region | City | Street | Mean (km) |
|---------|-------|-------|---------|--------|------|--------|-----------|
| im2gps3k | GeoCLIP | 59.8 | 39.4 | 20.6 | 16.6 | 7.7 | 2991.8 |
| | PIGEON* | 45.6 | 21.8 | 8.5 | 6.0 | 1.3 | 5220.1 |
| | GeoCLIP* | **71.3** | **44.0** | **26.0** | **17.2** | **7.2** | **2682.2** |
| yfcc4k | GeoCLIP | 63.4 | 38.3 | 17.2 | 8.9 | 4.1 | 3253.7 |
| | PIGEON* | 38.2 | 16.0 | 5.7 | 2.8 | 0.8 | 5035.3 |
| | GeoCLIP* | **67.3** | **40.4** | **18.6** | **9.6** | **3.7** | **2982.2** |
| yfcc26k | GeoCLIP | 62.2 | 37.6 | 17.4 | 9.9 | 4.4 | 3343.4 |
| | PIGEON* | 58.0 | 32.0 | 14.0 | 7.5 | 2.5 | 3850.0 |
| | GeoCLIP* | **66.7** | **41.9** | **19.8** | **10.1** | **4.2** | **2965.5** |

## 7 ETHICS STATEMENT

Indoor geolocation raises important considerations related to privacy and potential misuse. Our work focuses exclusively on understanding the feasibility of geolocation from publicly available images and does not attempt to identify individuals, private residences, or sensitive facilities. INDOOR-3.6M is constructed entirely from publicly available images that already include geographic metadata.

We acknowledge that geolocation technologies carry risks of misuse for surveillance or unauthorized location inference. We release this dataset strictly for research purposes. Any misuse for unauthorized surveillance, privacy-invasive applications, or identity inference is explicitly prohibited and strongly discouraged. Researchers must handle the data responsibly throughout algorithm development and when deploying public-facing technologies. To support privacy-preserving workflows, we provide SAM and YOLO annotations that enable downstream face and person masking. We expressly discourage any attempt at identity inference from this dataset.

Finally, our dataset may still reflect geographic, socioeconomic, and cultural biases present in public image corpora. We encourage future work to examine fairness, representation, and distributional shift in indoor geolocation models and to explore safeguards to prevent harmful or privacy-violating applications.

## 8 CONCLUSION AND FUTURE DIRECTIONS

Indoor scenes exhibit high intra-class variation, lack distinctive landmarks, and are subject to temporal change from renovations and redesigns. Promising directions include: (1) object-centric models leveraging culturally distinctive artifacts, (2) multi-scale reasoning combining global priors with fine-grained cues, and (3) robust methods addressing temporal drift. INDOOR-40K thus presents a challenging and scientifically rich testbed for advancing geolocation research.

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

# A    SUPPLEMENTARY MATERIALS

## A.1    INDOOR-40KDISTRIBUTIONS

Figure A1a captures the outdoor likelihood distributions of current geolocation benchmarks showing current benchmark datasets are dominated by outdoor scenes and are not suitable for evaluating indoor specific geolocation models. On the other hand Figure A1b demonstrates that INDOOR-40K contains mostly indoor images with a few ambiguous images in with p between 0.5 and 0.7. Figure A1c and A1d shows the scene and continent distribution in INDOOR-40K. The dataset exhibits diverse coverage across indoor scene categories, with the most represented scenes being bathrooms, bedrooms, restaurants, and various commercial spaces. The geographic distribution shows reasonable global coverage with Europe (33.4%), Asia (25.9%), and Africa (17.6%) being the most represented continents. Figure A1e illustrates the source distribution across INDOOR-3.6M and INDOOR-40Kdatasets. Flickr comprises the majority of both splits (56.5% and 69.3% respectively), followed by Booking.com (39.4% and 27.1%), with Wikipedia contributing a smaller portion (4.1% and 3.5%).

## A.2    RANDOM FOREST HYPERPARAMETERS, COUNTRY STATISTICS AND MODEL VALIDATION

To enable reproducibility of the diversity–prediction model, we report all Random Forest settings used in our experiments. We trained a RandomForestRegressor on two features—country population and land area—using 500 trees, random_state=42, max_depth=None, min_samples_split=2, min_samples_leaf=1, and the scikit-learn default for max_features. 5-fold cross-validation was used. Population and land-area values were sourced from publicly available World Bank datasets`https://data.worldbank.org/indicator/SP.POP.TOTL`, `https://data.worldbank.org/indicator/AG.LND.TOTL.K2`. Predictions from this trained model were used in all subsequent sampling stages. To justify the choice of this model, we compared its ability to predict country-level diversity scores against linear and quadratic baselines. Linear and polynomial (degree-2) regressions explain almost none of the variance ($R^2$ = 0.039 and 0.051 respectively), indicating that demographic factors relate to visual diversity in a highly non-linear way. In contrast, the Random Forest achieves $R^2$ = 0.861, showing that it captures this non-linear structure effectively and providing empirical justification for using its predictions as diversity-aware sampling weights in our dataset construction.

## A.3    GEOCLIP* ERROR ANALYSIS

Figure A2 offers a consolidated view of the factors that shape indoor geolocation performance. The loss distribution in Fig. A2a exhibits a clear long-tail pattern: while many predictions fall within a reasonable error range, a nontrivial portion deviate substantially, underscoring the uneven difficulty across indoor scenes. Regional patterns in Fig. A2b further illustrate this variability, with some continents showing consistently higher mean error than others—suggesting that geographic context, and possibly regional imbalance in scene characteristics, plays a role in model performance. The relationship between confidence and accuracy in Fig. A2c shows that the model's indoor-confidence score is a meaningful indicator of reliability, with higher confidence generally corresponding to improved accuracy and lower median error. Finally, Fig. A2d highlights pronounced differences across scene categories: visually distinctive environments achieve substantially lower errors, whereas more generic or repetitive scenes remain challenging.

## A.4    ABLATION ON INDOOR-LIKELIHOOD THRESHOLD

To assess whether low-confidence indoor scenes introduce noise, we retrain GeoCLIP* on subsets of INDOOR-3.6M defined by strict indoor-likelihood thresholds $P_{\min} > \{0.5, 0.6, 0.7, 0.8, 0.9\}$ and evaluate all models on the full INDOOR-40K benchmark. As shown in Table 6, performance is highly stable for thresholds up to $P_{\min} > 0.7$, with negligible variation in accuracy across spatial scales and mean geodesic error. Fine-grained performance (25 km and 1 km) also remains unchanged, indicating that ambiguous indoor scenes ($0.5 \leq P < 0.7$) do not harm geolocation. In contrast, stricter filtering ($P_{\min} > 0.8$ or $> 0.9$) consistently degrades accuracy and increases mean

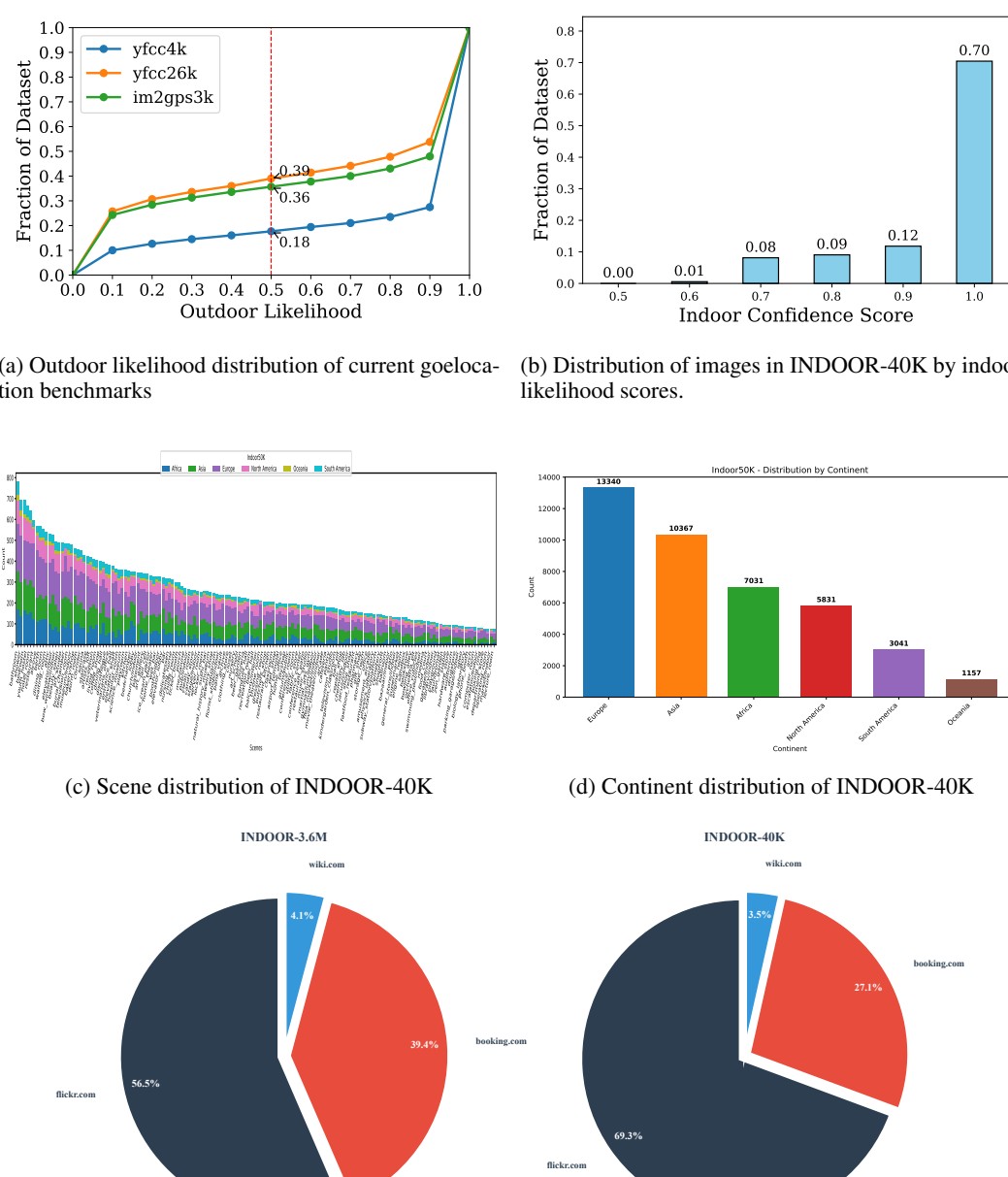

(a) Outdoor likelihood distribution of current goelocation benchmarks

(b) Distribution of images in INDOOR-40K by indoor likelihood scores.

(c) Scene distribution of INDOOR-40K

(d) Continent distribution of INDOOR-40K

(e) Data source distribution in training and test splits.

Figure A1: Comprehensive analysis of INDOOR-40K dataset composition. (a) Comparison with existing benchmarks shows outdoor scene bias. (b) Indoor confidence score distribution demonstrates more images depict completely indoor scenes. (c) Scene categories span diverse indoor environments. (d) Geographic distribution across continents. (e) Data sources are similarly balanced across training and test splits.

error from 3178 km to 3282 km and Acc@1 km dropping from 3.77% to 2.85%. These results show that borderline indoor images are informative rather than noisy, and retaining all samples with $P \geq 0.5$ yields the best overall performance.

Table 6: Ablation on indoor-likelihood threshold $P_{\min}$ applied to the INDOOR-3.6M training set. GeoCLIP* is retrained on each subset and evaluated on INDOOR-40K .

| $P_{\min}$ | 2500 km (%) | 750 km (%) | 200 km (%) | 25 km (%) | 1 km (%) | Mean (km) |
|---|---|---|---|---|---|---|
| All | **66.1** | **40.3** | **22.3** | **11.5** | **3.8** | **3178** |
| > 0.5 | 66.1 | 40.3 | 22.0 | 11.1 | 3.7 | 3231 |
| > 0.6 | 65.9 | 39.9 | 21.6 | 11.0 | 3.5 | 3249 |
| > 0.7 | 66.0 | 39.9 | 21.2 | 10.6 | 3.5 | 3210 |
| > 0.8 | 65.7 | 39.1 | 20.9 | 10.7 | 3.4 | 3262 |
| > 0.9 | 65.4 | 38.5 | 19.8 | 9.5 | 2.9 | 3282 |

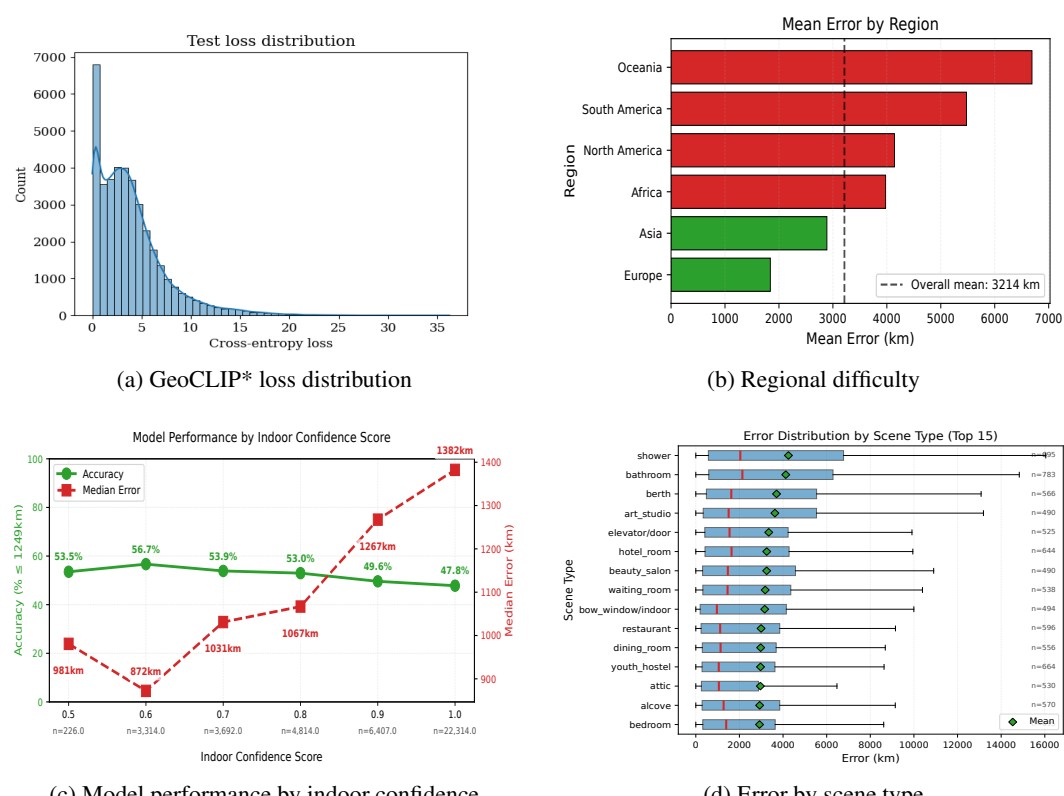

(a) GeoCLIP* loss distribution

(b) Regional difficulty

(c) Model performance by indoor confidence

(d) Error by scene type

Figure A2: Analysis of indoor geolocation difficulty across four dimensions: (a) Overall loss distribution, (b) Regional variation in mean error, (c) Performance as a function of indoor-confidence score, and (d) Error variation across scene categories.

## A.5 CLIP vs GeoCLIP* Embedding Analysis

We analyze clustering behavior across three embedding spaces: (i) Pretrained CLIP (ViT-L/14) embeddings, trained for generic vision-language alignment; (ii) GeoCLIP* image embeddings (Geo-CLIP*), which incorporate geographic supervision into indoor image features; and (iii) GeoCLIP* location embeddings, which embed GPS coordinates into the same latent space as the indoor images. We apply hierarchical agglomerative clustering to the test set embeddings for each embedding type. Figures A3 and A4 illustrate sampled images from resulting clusters with country and continent labels. To examine global structure, we also project all images onto a world map and color them by cluster assignment. This allows direct inspection of whether clusters correspond to coherent geographic regions. We observe that Pretrained CLIP embeddings(i) produce clusters that reflect semantic similarity but are geographically incoherent, while GeoCLIP* image embeddings(ii) show improved regional consistency, with clusters more often localized to specific continents. GeoCLIP* location embeddings(iii), which are derived from GPS coordinates rather than images, exhibit the

strongest geographic alignment, with clusters corresponding to contiguous world regions. This suggests that the joint embedding space effectively bridges image and location representations, enabling geographically coherent structure to emerge.

### A.6 VISION-LANGUAGE MODEL PROMPTING

We evaluate VLMs using zero-shot prompting with structured text queries. For each image, we issue two separate prompts requesting geographic predictions at continent and country granularity:

**Continent-level prompt:**

```
Given the following indoor image, estimate the continent
where the photo was taken.  Respond with the single most
likely continent name selected from this list:  [Africa,
Asia, Europe, North America, South America, Oceania].
Answer with only the continent name.
```

**Country-level prompt:**

```
Given the following indoor image, estimate the country where
the photo was taken.  Respond with the single most likely
country name selected from this list:  [223 countries].
Answer with only the country name.
```

The country list includes all countries represented in INDOOR-40K. Both prompts enforce constrained outputs to enable automated evaluation. Models receive no additional context or examples beyond the image and prompt.

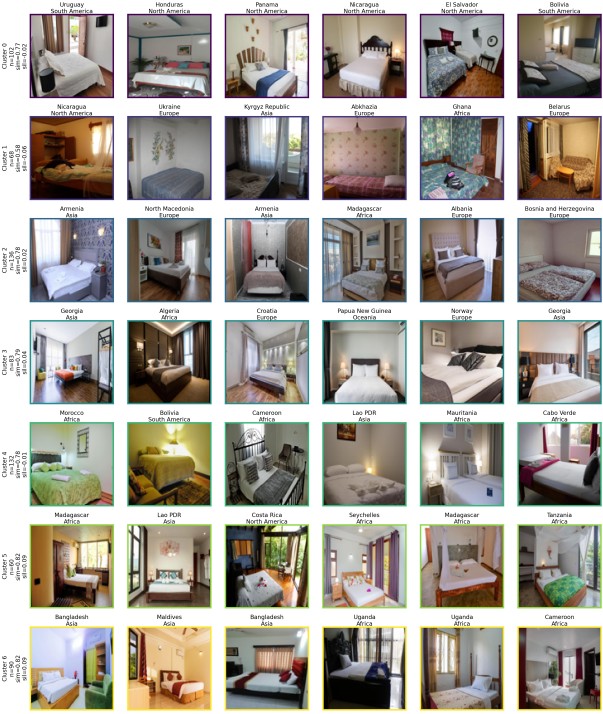

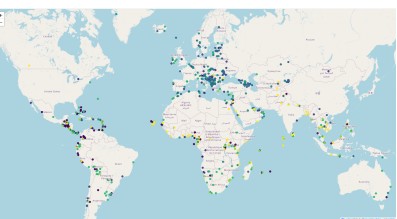

(a) CLIP bedroom image clusters          (b) CLIP clusters on world map

Figure A3: **Pretrained CLIP (ViT-L/14) Image embeddings:** Clusters are semantically coherent (a) but geographically incoherent (b). Images with similar layouts or styles appear grouped together, yet cluster assignments scatter across multiple continents.

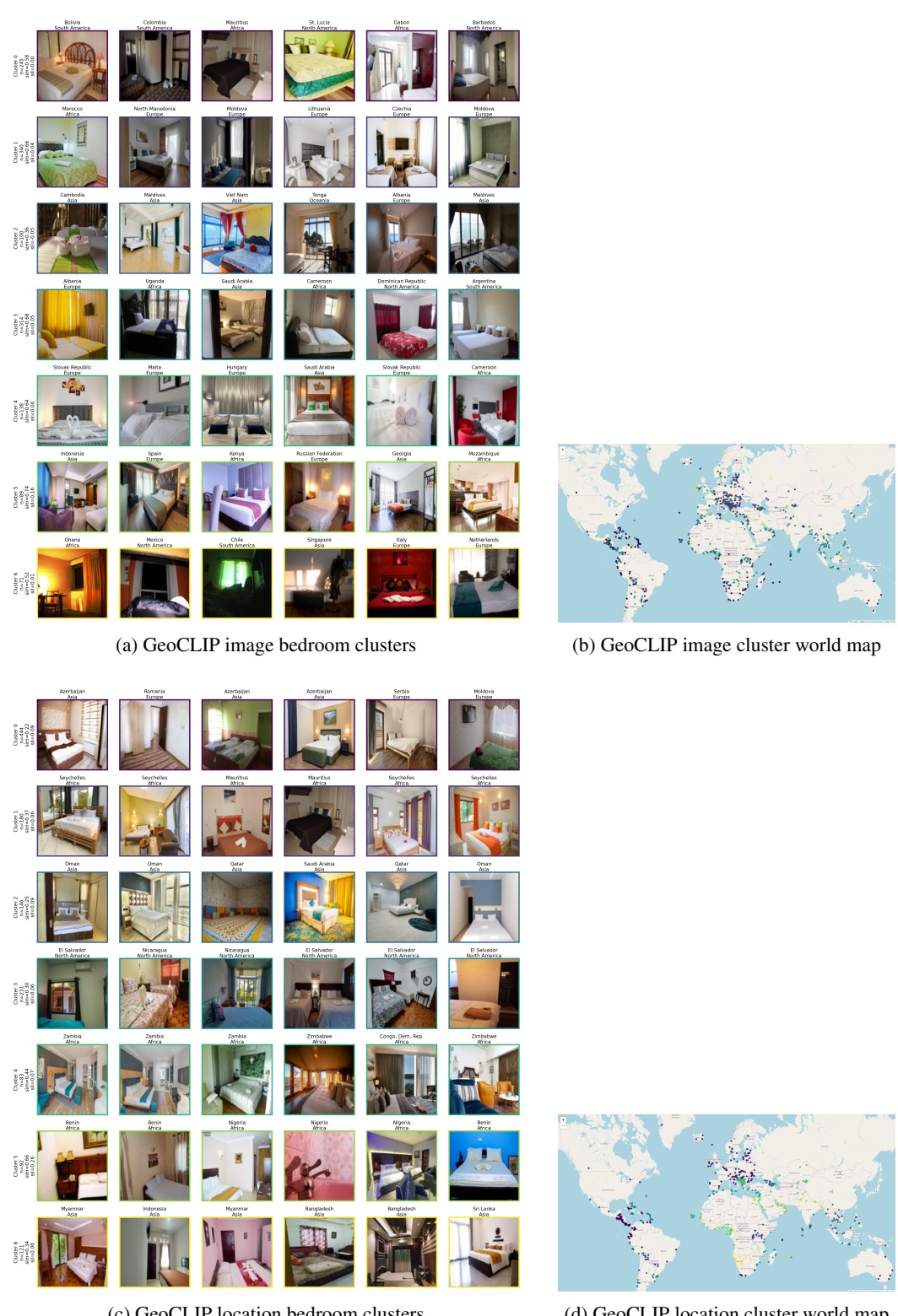

(a) GeoCLIP image bedroom clusters

(b) GeoCLIP image cluster world map

(c) GeoCLIP location bedroom clusters

(d) GeoCLIP location cluster world map

Figure A4: **GeoCLIP embedding analysis.** Top: Image embeddings show improved continent- and region-level grouping compared to pretrained CLIP. Bottom: Location embeddings exhibit the strongest geographic coherence, with clusters forming contiguous world regions.

