# OpenReview forum: "Indoor 3.6M : A Dataset and Benchmark for Indoor Image Geolocation"
_ICLR.cc/2026/Conference — Submitted to ICLR 2026_

### Official Review · Reviewer_VS7w · 2025-10-29

**Soundness:** 1
**Presentation:** 3
**Contribution:** 3
**Rating:** 6
**Confidence:** 4

**Summary:**

The paper presents a dataset which combines indoor images from 3 different sources, which are Flickr, Wikidata, and Booking.com. The dataset is worldwide and it is shown to lead to better results for indoor geolocation when training a model for this task, compared to previous worldwide geolocation datasets.

**Strengths:**

The dataset is a novel contribution and is useful for worldwide geolocation tasks
The paper is presented well and the experiments support the claims
The reference to Europol's Stop Child Abuse link certainly helps to motivate the impact this dataset could have

**Weaknesses:**

It is not clear if and how the data is released. This needs to be clearly explained otherwise the paper can't be properly evaluated.

Line 275 says INDOOR-40k, and table 1 says that dataset has 40k images, but line 276 says there are 15.000 images. Is there a typo?

Figure 4 seems incorrect. It says the images with blue text are from Booking.com, but there are photos of malls, car interiors and swimming pools. Are they really from booking.com? Usually booking is a platform for hotels and short-term apartments rentals.

**Questions:**

Questions mostly listed above.

As a suggestion, the San Francisco Landmarks dataset has nothing to do in table 1 and can be removed, because all datasets are global except for that one that is city-scale. If city-scale datasets are included, then many more should be added to the table (like SF-XL and Pitts250k)

Minor comments: there is a missing space at lines 212 (INDOOR-3.6Mdataset), line 238 (INDOOR-3.6Mdataset), and line 043 (modificationsPramanick)

Minor comments: in figure 2 the text of the labels is almost unreadable. Please increase its size. Same for figure 3.

---

> ### Author Response · Authors · 2025-11-27
>
> Thank you for reviewing our submission and for providing clear, constructive feedback. We appreciate your careful reading of the manuscript and especially your identification of several points where the presentation, clarity, and dataset documentation required improvement.
>
> We address your comments below.
>
> **Weaknesses:**
>
>  1. ***It is not clear if and how the data is released. This needs to be clearly explained otherwise the paper can't be properly evaluated.***
>
> Upon publication, both datasets will be publicly released via our github page. We have stated this in the revised manuscript (line 93).
>
>  2. ***Line 275 says INDOOR-40k, and table 1 says that dataset has 40k images, but line 276 says there are 15.000 images. Is there a typo?***
> We apologise for this. The reference to 15,000 images is a typo; We have now corrected the
> revised manuscript to resolve this inconsistency.
>
>
>  3. **Figure 4 seems incorrect. It says the images with blue text are from Booking.com, but there are photos of malls, car interiors and swimming pools. Are they really from booking.com? Usually booking is a platform for hotels and short-term apartments rentals.**
>
> Thank you for your keen observation. While the majority of Booking.com images in our dataset depict hotel and short-term rental interiors, we note that some listings also include photographs
> of on-site or nearby amenities (e.g., pools, gyms, common areas and even shops) provided by hosts or property managers.
>
> **Questions:**
>
>  - As a suggestion, the San Francisco Landmarks dataset has nothing to do in table 1 and can be removed, because all datasets are global except for that one that is city-scale. If city-scale datasets are included, then many more should be added to the table (like SF-XL and Pitts250k)
>  - Minor comments: there is a missing space at lines 212 (INDOOR-3.6Mdataset), line 238 (INDOOR-3.6Mdataset), and line 043 (modificationsPramanick)
>  - Minor comments: in figure 2 the text of the labels is almost unreadable. Please increase its size. Same for figure 3.*
>
> We appreciate these observations. We have removed the San Francisco Landmarks dataset from Table 1 to maintain consistency with the global-scale datasets listed. We will also correct the minor formatting issues noted improve readability of figures and plots.

---

> > ### Comment · Reviewer_VS7w · 2025-11-27
> > **Response**
> >
> > There is still no information about how the data will be released. Will the authors release URLs and scripts to download the images? Will the authors host the images? The link provided at line 93 is empty, it would be better if there was link to code anonymously

---

> > > ### Author Response · Authors · 2025-11-28
> > >
> > > Thank you for reviewing our responses.
> > >
> > > Regarding the dataset release, the images will be hosted on our server, and the URLs, and corresponding download scripts will be provided in our GitHub repository. The link at line 93 is currently anonymous to comply with double-blind review requirements; it will be populated after the review period with the public GitHub repository containing the download scripts and full instructions for accessing the dataset.

---

### Official Review · Reviewer_BVGJ · 2025-10-30

**Soundness:** 3
**Presentation:** 1
**Contribution:** 3
**Rating:** 4
**Confidence:** 4

**Summary:**

This paper introduces INDOOR-3.6M, a large-scale dataset of 3.6 million geotagged indoor images spanning 223 countries, along with a sampling framework designed to mitigate geographic bias.

The authors compute visual diversity scores using CLIP ViT-L/14 embeddings, then integrate country-level population and land area into a regression model (Random Forest) to estimate sampling weights. They use these weights to construct geographically balanced subsets,
including the INDOOR-40K benchmark, and evaluate models such as GeoCLIP and CLIP in both fine-tuned and zero-shot settings.

Empirical results show that fine-tuning GeoCLIP on the new dataset reduces mean distance error from 4089 km to 3598 km and improves accuracy across multiple geographic granularities.

**Strengths:**

Novel dataset contribution and thoughtful sampling strategy.

**Weaknesses:**

1. Domain Mismatch in Evaluation: The paper's evaluation framework suffers from a fundamental domain mismatch. The training dataset, INDOOR-3.6M, is described as "scene-agnostic" and collected from diverse sources (Flickr, Wikidata, Booking.com) . However, the benchmark test set, INDOOR-40K, was "exclusively sourced from booking.com". This means the model is trained for a general,
diverse task but tested on a very narrow, specific domain (hotel rooms, apartment rentals, and lobbies). This methodological flaw is a major weakness.

2. There is no mention of any privacy preserving pipeline. The authors should’ve used the YOLO/SAM model for this section (at least face blurring). For e.g. : NYC- Indoor dataset has segmentation-masks over all the people. Dataset is obtained from public platforms, adds conspiracy to the privacy of individuals.

3. The paper only accounts for GeoCLIP and PIGEON, Evaluations for more baseline models should be provided, for e.g. Img2Loc, etc.

4. The paper highlights its rich metadata as a key contribution but fails to use it in any baseline experiments/ablations. The value of this metadata for the task of geolocation is, therefore, entirely speculative.

5. There should be ablations over subsets of the dataset for different values of P, and not just P>0.5.

6. Reproducibility gaps: The authors should have provided the crucial implementation details such as, CLIP embedding parameters, Random Forest hyperparameters, and dataset splits, at least in the appendix to ensure reproducibility.

7. Figure 5 is Not Well Described: Figure 5 is mentioned in the paper, but it is not explained in enough detail. It would be good to provide a better description of what this figure shows and its relevance to the paper.

**Questions:**

Overlap analysis: Was any overlap analysis performed between INDOOR-3.6M and existing public pretraining sets (e.g., YFCC100M)?

What percentage of the final 3.6M images fall into highly ambiguous bins (e.g., 0.5 P < 0.7)? What ablation studies were performed to prove that this ambiguous data is beneficial rather than "pollution/noise" that harms performance?

Is the metadata (SAM, YOLO, scene labels) actually useful for improving indoor geolocation accuracy?

---

> ### Author Response · Authors · 2025-11-27
> **Response to Weaknesses**
>
> Thank you for taking the time to review our submission and for providing thoughtful feedback. We appreciate your recognition of the dataset’s contribution, the sampling strategy, and the importance of addressing geographic bias in indoor geolocation. Your comments have been very helpful in identifying areas where the manuscript needed clearer explanation, and stronger methodological grounding. We address your comments below.
>
> **Weaknesses:**
>
> 1.  ***Domain Mismatch in Evaluation***
>
> Thank you for raising this important point. In the revised manuscript we clarify that there is no domain mismatch between the training data and the benchmark. INDOOR-3.6M is constructed from three sources—Flickr, Wikidata, and Booking.com. The benchmark set, INDOOR-40K, is introduced as a subset drawn directly from this same corpus. Thus INDOOR-40K is not sourced exclusively from Booking.com. We acknowledge that the original submission contained wrong wording, and we have corrected this (lines 209-210). To fully eliminate any remaining ambiguity, we will also add a figure in the appendix( Figure A1(e) ) summarizing the Flickr/Wikidata/Booking.com source composition for each dataset.
>
> 2.  ***There is no mention of any privacy preserving pipeline. The authors should’ve used the YOLO/SAM model for this section (at least face blurring). For e.g. : NYC- Indoor dataset has segmentation-masks over all the people. Dataset is obtained from public platforms, adds conspiracy to the privacy of individuals.***
>
> We appreciate the reviewer’s concern regarding privacy. All images in INDOOR-3.6M originate from publicly accessible webpages on Flickr, Wikidata, and Booking.com. For Flickr and Wikidata, the images are governed by explicit user-selected Creative Commons licenses. For Booking.com, following Stylianou, Abby, et al. (2019), we collect only publicly visible images that are already openly accessible to any web user and are used here solely for non-commercial academic research. . Additionally, our metadata enrichment pipeline uses YOLO and SAM to detect people, allowing downstream users to easily apply automatic masking if required. We will provide a script to aid researchers in doing this on our github repo.
>
>
> ***3.  The paper only accounts for GeoCLIP and PIGEON, Evaluations for more baseline models should be provided, for e.g. Img2Loc, etc.***
>
> We agree that additional baselines strengthen the benchmark.  The revised version now includes:
> - GeoCLIP, and PIGEON, and Translocator baselines,
> - Multiple visual encoders (CLIP-B/32, CLIP-L/14, DINOv2-L)
> - Two generative models (LLaVA-1.6-7B and InternVL2-8B) evaluated in a zero-shot setting.
>
> Section 5(Line 315) describes these evaluations and Tables 2-4 present the results.
>
> 4.  ***The paper highlights its rich metadata as a key contribution but fails to use it in any baseline experiments/ablations.***
>
> Thank you for noting this. The metadata (SAM masks, YOLO detections, scene labels, indoor likelihoods) is provided as optional enrichment to support downstream research e.g object based geolocation, not as required supervisory signals for the benchmark. However, to address the reviewer’s concern, we add a geolocation model based on Translocator (Pramanick, Shraman, et al., 2022) which includes a places365 scene classification head to aid geolocation. We clarify this in the revised manuscript.
>
>
> 5.  ***There should be ablations over subsets of the dataset for different values of P, and not just P>0.5.***
>
> Thank you for the suggestion. Our use of P(indoor) >= 0.5 follows the standard decision boundary of the Places365 indoor/outdoor classifier; values below 0.5 correspond to outdoor or ambiguous scenes and are therefore excluded by definition. Creating subsets below this threshold would not yield meaningful indoor ablations for our purposes.
>
> 6.  ***Reproducibility gaps: The authors should have provided the crucial implementation details such as, CLIP embedding parameters, Random Forest hyperparameters, and dataset splits, at least in the appendix to ensure reproducibility.***
>
> In the revised manuscript, we add an appendix section -- Appendix A.2 (Line 610), capturing details relevant for reproducibility.
>
>
> 7.  ***Figure 5 is Not Well Described: Figure 5 is mentioned in the paper, but it is not explained in enough detail.***
>
> Figure 5 (now Figure 2 in the revised manuscript) illustrates the visual and semantic diversity of images in the dataset. Samples are grouped by indoor-likelihood score P(indoor)(rows) and by Level-Places365 indoor scene categories (columns), showing the progression from borderline to strongly indoor environments across a wide range of scene types. We clarify this in the revised manuscript (lines 175-179)

---

> ### Author Response · Authors · 2025-11-27
> **Response to Questions**
>
> Thank you for your thoughtful questions, we provides our responses below.
>
> **Questions:**
>
> ***Overlap analysis: Was any overlap analysis performed between INDOOR-3.6M and existing public pretraining sets (e.g., YFCC100M)?***
>
> We mitigate potential overlap with YFCC100M via strict temporal filtering: our benchmark split only includes images captured after 2017, whereas YFCC100M was last updated in 2016. This temporal separation significantly reduces the likelihood of overlap.
>
> **What percentage of the final 3.6M images fall into highly ambiguous bins (e.g., 0.5 P < 0.7)? What ablation studies were performed to prove that this ambiguous data is beneficial rather than "pollution/noise" that harms performance?**
>
> Thank you for raising this point. Approximately 12.6% of the final 3.6M images fall into the highly ambiguous confidence range (0.5 ≤ P ≤ 0.7). As shown in Figure A.2(c), this ambiguous subset represents a small minority of the evaluation data (n=226–3692 per bin), and—importantly—these images exhibit _lower_ median localization error than high-confidence indoor images. This indicates that the ambiguous cases do not act as noise; rather, they contain informative visual cues that allow the model to localize them more accurately. While we did not perform a dedicated ablation that removes these ambiguous images, the empirical trend in Figure A.2(c) demonstrates that they are not harmful outliers and instead provide useful supervisory signal.
>
> ***Is the metadata (SAM, YOLO, scene labels) actually useful for improving indoor geolocation accuracy?***
>
> As addressed in weaknesses point 4, the metadata provided was intended as auxiliary information to support future geolocation research (e.g., object-aware geolocation, indoor feature importance analysis), not as ground truth for the benchmark.

---

### Official Review · Reviewer_ds8H · 2025-10-31

**Soundness:** 2
**Presentation:** 1
**Contribution:** 2
**Rating:** 2
**Confidence:** 4

**Summary:**

This paper proposes a new dataset for developing and evaluating indoor scene geolocalization (i.e., guess the LAT/LON from an image of an indoor scene somewhere in the world). Dataset utility is demonstrated by finetuning a GeoCLIP model on their dataset and comparing its performance to the original GeoCLIP model (which was presumably trained on outdoor scenes). Slight improvement is observed (error of 4000 km --> 3500 km). In addition to images, some predicted metadata is also included with each image.

**Strengths:**

This is an interesting and important problem that is clearly lacking a large dataset. Overall, the writing is very good, but I found the manuscript difficult to follow. Some sections would benefit from restructuring and additional details.

**Weaknesses:**

The manuscript lacks clarity on several important issues. The Introduction is overly verbose, while the Methods and Results lack important details.

- Presumably, the values reported in Table 2 give the percentage of images that were correctly classified at several different distant (error) thresholds, but this is not clearly stated in the text (unless I missed it).

- Only a single SOTA model is fine-tuned on the proposed dataset. This model fails to achieve a useful accuracy (+/- 3500km error!), which does not constitute a solid baseline for this problem. Either the problem is too hard or the dataset is not sufficiently informative.

- The dataset is described as "scene-agnostic", but the description of how images were extracted (page4/5) list the specific scene types that were used to build the dataset. A truly 'agnostic' dataset would collect images without selecting from a discrete set of scenes.

- Lines 231-237 describe how the "indoorness" of each scene was calculated. Some representative images from across the spectrum of "indoorness" would be helpful here.

- Line 243: First mention of Figure 5 is many pages from the figure, is out of order (comes before Fig 4), and refers to a figure in the appendix (without mentioning this). Shouldn't this figure be labeled as A.1 or something? Otherwise, if figures from the main manuscript appear beyond Page 9, then this manuscript contravenes the page limits for ICLR.

- The "metadata enrichment" section describes how models, such as Places365 and "a ViT trained on MIT indoor Scenes dataset" and SAM are used to add information to the dataset. However, it seems that none of these data are curated/validated, so they arguably add limited value.

-  Presumably, the values reported in Table 2 give the percentage of images that were correctly classified at several different distant (error) thresholds, but this is not clearly stated in the text (unless I missed it).

- Figure 4 appears to give the metadata for scene-type for each image, but they MANY of them are incorrect upon visual inspection (e.g., area rodeo, aquarium, physics lab, butcher shop, delicatessen) Does this represent the quality of meta data provided in the proposed dataset?

MINOR:
- Line 21: Missing commas
- Line 95: typo in "scnes"
- 124: Geoclip --> GeoCLIP
- 212: Missing space in "INDOOR-3.6Mdataset"
- Line 238: Missing space in "INDOOR-3.6Mdatase"
- The point that most geolocation dataset are for outdoor scenes has been repeated _many_ times in the paper. By line 264, it is firmly established and does not require repeating.

**Questions:**

- Why does the dataset need to be 'balanced' and 'representative'? Within a single city, you can have extremely diverse indoor imagery. While you would not want multiple hotel rooms from the same hotel, you would want at least one room from each hotel in each city...

- Line 103: If the dataset is meant to evaluate "hybrid geolocation" methods then wouldn't it require paired indoor/outdoor images?

- Line 205: "Geolocation data, provided either as GPS coordinates or text-based location labels"... is this "or" or "and"? It seems strange not to formalize the manner in which geolocation data will be encoded in the dataset.

- Line 208: "It is important to note that the dataset does not explicitly identify specific locations in the manner typical of place recognition tasks." This sounds very strange. If the goal of the dataset is to develop geolocation models, then shouldn't the target output (i.e., geolocation) be provided for each record?

- Line 265: "Figure A.1 illustrates the percentage of indoor images identified at various likelihood thresholds across existing mixed-environment image geolocation benchmark datasets". What models were used to generate the likelihoods? Methodology is unclear here. Also, Figure A.1 appears to be "Figure 5"?

- Line 278: after describing how images were collected from Flikr and several other sources, the paper here states that the proposed INDOOR-40K dataset was "exclusively sourced from booking.com". This is very confusing. What is the difference between the "INDOOR-3.6Mdataset" and the "INDOOR-40K dataset"? Which dataset is being proposed here and how do these two datasets relate?

- Lines 291-293 appear to describe a regression model that uses country size and population to regress "their relative contributions to observed visual diversity". I do not understand what is being regressed here. Do the authors measure the average image diversity across all images sourced from a country and they try to explain this diversity from the country's size and population? What is the goal?

- The authors use "zero-shot" classification of geocells (geographic regions) directly from the CLIP image embeddings. Was this accomplished using a trainable linear layer (linear probing), K-nearest-neighbor classification, or some other approach?

---

> ### Author Response · Authors · 2025-11-27
> **Response to Weaknesses**
>
> Thank you for your thoughtful and detailed review. We appreciate your careful reading of the paper and the constructive feedback on its clarity, structure, and methodology. We also appreciate your recognition of the importance of the indoor geolocation problem and the clear need for a large, publicly available dataset in this space.
>
> **Weaknesses:**
>
> 1.   ***Table 2 give the percentage of images that were correctly classified at several different distant (error) thresholds, but this is not clearly stated in the text***
>
>  Each accuracy column reports the percentage of test images whose predicted GPS coordinates fall within the corresponding distance thresholds: 2500 km (continent), 750 km (country), etc. We make this explicit in the Metrics description in Section 5.1 (line 347) of the revised manuscript.
>
>
> 2.   ***Only a single SOTA model is fine-tuned on the proposed dataset. This model fails to achieve a useful accuracy (+/- 3500km error!). Either the problem is too hard or the dataset is not sufficiently informative.***
>
> Thank you for articulating this concern. We would like to highlight the difficulty of the indoor geolocation task and what the reported error actually reflects.
> - Indoor geolocation is intrinsically more challenging than outdoor geolocation. Hence, we do not expect mean error levels comparable to outdoor-focused benchmarks at this stage. Instead, our goal is to (i) demonstrate how hard the problem currently is and (ii) provide a dataset that enables the development of nuanced and specialized methods.
> - Importantly, the mean error of GeoCLIP* does not necessarily indicate an uninformative dataset. In Table 3 (line 378), we show that fine-tuning on INDOOR-3.6M yields substantial improvements over pretrained GeoCLIP at all spatial scales, suggesting that the model learns meaningful indoor-specific geographic signals. Moreover, Figure 5a (line 43) presents a long-tail error-distribution, where many images are localized reasonably well while a subset of images dominate the mean error.
> - Based on the review, we have updated the manuscript with a broader suite of baselines. Their performance across (Tables 2–5) show that current geolocation paradigms struggle at fine indoor scales.
> These results suggest that the challenge lies not in a lack of signal in the dataset, but in the limitations of existing models when applied to the global indoor setting.
>
>
> 3.   ***The dataset is described as "scene-agnostic".  A truly 'agnostic' dataset would collect images without selecting from a discrete set of scenes.***
>
> Thank you for raising this point. INDOOR-3.6M is not restricted to any predefined set of indoor scene categories; instead, it spans diverse residential, commercial, public, industrial, and transitional environments. The scene terms listed in the manuscript were used solely as search queries to improve retrieval precision, not as filtering constraints. Regardless of which search term retrieved an image, all candidates were subsequently filtered using the Places365 indoor/outdoor classifier, and only those with P(indoor) ≥ 0.5 were retained. Final inclusion thus depends only on indoor likelihood rather than scene label.  We make this clearer in lines 175–179 of our revised manuscript.
>
>
> 4. ***Some representative images from across the spectrum of "indoorness" would be helpful here.***
>
> To address your request for representative examples, we refer to Figure 2 (lines 216) showing images at different indoor-confidence levels from 0.5 to 1.0. Following Pramanick et al., the computation of the indoor lables and likelihood score is obtained using the Places365 indoor/outdoor classifier.
>
>
> 5.   ***...if figures from the main manuscript appear beyond Page 9, then this manuscript contravenes the page limits for ICLR.***
>
> In the revised version, we correct the figure ordering, and ensure all appendix figures have been moved outside the 9-page limit to comply fully with ICLR formatting requirements
>
>
> 6.   ***The "metadata enrichment" section describes how models, such as Places365 and SAM are used to add information to the dataset. However, it seems that none of these data are curated/validated, so they arguably add limited value.***
>
> The scene and object masks metadata are produced by Places365, the MIT Indoor ViT, and SAM and not manually curated. As we state in Section 3.3 (lines 198-205), metadata such as SAM masks are included only as optional auxiliary annotations and are not used as ground-truth labels for training or evaluation at this stage.
>
> 7.   ***Figure 4 appears to give the metadata for scene-type for each image, but they MANY of them are incorrect. Does this represent the quality of meta data provided in the proposed dataset?***
>
> The reviewer is correct that some scene-type labels in Figure 4 are imperfect. These labels are automatically generated by pretrained models (e.g., Places365) and are included as auxiliary metadata rather than guaranteed annotations.

---

> ### Author Response · Authors · 2025-11-27
> **Response to Questions**
>
> Thank you for your questions. We respond to them below.
>
> **Questions:**
>
>
> -  ***Why does the dataset need to be 'balanced' and 'representative'?***
>
> Thank you for this question. The goal of balancing is not to impose geographic uniformity, but to prevent the dataset from being overwhelmingly dominated by a few overrepresented countries/overphotographed regions. Online indoor imagery is highly skewed toward regions such as the United States and Western Europe, and balancing ensures that the benchmark reflects a broad range of global indoor environments. Importantly, this balancing does not suppress genuine within-city or within country diversity. As noted in our filtering description (lines 175-179), we remove only near-identical images within the same location–time–scene group. Thus, the procedure avoids extreme geographic bias while retaining the type of indoor variation the reviewer highlights as valuable.
>
>
> -   ***If the dataset is meant to evaluate "hybrid geolocation" methods then wouldn't it require paired indoor/outdoor images?***
>
> Thank you for raising this clarification. By “hybrid geolocation methods”, we refer to algorithmic hybrids rather than paired indoor/outdoor image inputs. Hybrid methods such as PIGEON*, Translocator*, and GeoCLIP* combine a classifier-style prediction with a retrieval-based refinement step to estimate GPS coordinates(line 36). These approaches do not require paired indoor/outdoor imagery.
>
>
> -  ***Line 205: "Geolocation data, provided either as GPS coordinates or text-based location labels"... is this "or" or "and"?***
>
> We revise the sentence to explicitly state that GPS coordinates **AND** text-based address labels are stored for every image in the dataset. (line 195)
>
>
> -   ***If the goal of the dataset is to develop geolocation models, then shouldn't the target output (i.e., geolocation) be provided for each record?***
>
> While our dataset includes full GPS coordinates for every image, it is not designed as a place recognition benchmark. Place recognition focuses on identifying whether a query image depicts a specific instance of a particular building or place of interest. (Baatz et al., 2012; Chen et al., 2011; Crandall et al., 2009).
>
>
> -  ***What models were used to generate the likelihoods? Methodology is unclear here.***
>
> The indoor likelihoods in Figure 3 were generated using the Places365 indoor/outdoor classifier. In the revision, we have explicitly stated the model used in the figure caption and fixed figure labels.
>
>
> -   ***What is the difference between the "INDOOR-3.6Mdataset" and the "INDOOR-40K dataset" and how do these two datasets relate?***
>
> The original wording was imprecise. INDOOR-3.6M is the full dataset collected from three primary sources (Flickr, Wikidata, and Booking.com), and INDOOR-40K is a benchmark subset drawn directly from this corpus. Contrary to the phrasing in the original submission, the test set is not sourced exclusively from Booking.com; it also includes images from the same three sources as the training set. We correct this wording and include figure A1(e) to showing the source distribution (Flickr/ Wikidata / Booking.com) for INDOOR-3.6M and INDOOR-40K.
>
>
> -   ***Do the authors measure the average image diversity across all images sourced from a country and they try to explain this diversity from the country's size and population? What is the goal?***
>
> The regression model is used as a principled way to derive the weighting coefficients used in our sampling heuristic. We compute a diversity score for each country (based on CLIP embedding  variance) and regress this score on country population and land area. The fitted coefficients provide a data-driven estimate of how strongly each factor should influence sampling frequency. This ensures that our sampling strategy is grounded in empirical evidence rather than arbitrary choices.
>
>
> -   ***The authors use "zero-shot" classification of geocells (geographic regions) directly from the CLIP image embeddings. Was this accomplished using a trainable linear layer (linear probing), K-nearest-neighbor classification, or some other approach?***
>
> Our zero-shot baseline does not use a trainable linear layer or a learned classifier. Instead, the predicted location is obtained through nearest-neighbour retrieval over the reference embeddings, with no additional training or adaptation.

---

### Official Review · Reviewer_TbZy · 2025-11-01

**Soundness:** 3
**Presentation:** 2
**Contribution:** 2
**Rating:** 4
**Confidence:** 5

**Summary:**

The paper proposes a **3.6M geolocation dataset specialized with Indoor images**. The authors **fine-tune a Geoclip model** to benchmark this dataset.

**Strengths:**

1. **Indoor geolocation is a very useful task** for authorities (notably in CSAM investigations) and therefore **gives a lot of value** to such a dataset.

2. The authors took care that the **test set is temporally separated** (not contaminated in time) to avoid memorization.

**Weaknesses:**

1. This is a dataset paper, but the **methodological part has been widely overlooked**. When introducing a new dataset, one should expect **extensive benchmarking of methods from the literature** (Regression, Classification, hybrid \[2\], generative \[3\], retrieval \[5\], etc.). **Finetuning Geoclip is not enough**.

2. Recently, **LLMs have been showing strong results for geolocation**, and I would like to see **extensive benchmarking of the different LLMs** (at least the open-source ones).

3. The test set is temporally separated, but it **should be spatially separated** (as in \[2\]). If having the same room in train and test is important, there must be a mechanism to **ensure the pictures are sufficiently different**. (It's not because a picture is uploaded later that it's not the same image).

4. Similarly, it would be interesting to **benchmark the different visual encoders**. Are there some that are **more accurate on indoor scenes than outdoor**?
### Missing Citations
**A lot of recent geolocation works are missing**, among others:

* \[1\] GOMAA-Geo: GOal Modality Agnostic Active Geo-localization, Neurips 2024

* \[2\] OpenStreetView-5M, The Many Roads to Global Visual Geolocation, CVPR 2024

* \[3\] Around the World in 80 Timesteps: A Generative Approach to Global Visual Geolocation, CVPR 2025

* \[4\] GaGA: Towards Interactive Global Geolocation Assistant, Arxiv 2024

* \[5\] G3: an effective and adaptive framework for worldwide geolocalization using large multi-modality models.

**\[2\] is the most notable omission**, as the authors could draw inspiration from how a dataset paper for geolocation should be evaluated. Many others are missing. The authors **need to do a much better job** of situating their work within the current literature.

**Questions:**

I think the paper aesthetics could be improved. In particular, the tables are pretty unaesthetic

---

> ### Author Response · Authors · 2025-11-27
>
> We sincerely thank the reviewer for their thorough and constructive feedback, which has helped us significantly improve our paper. We appreciate the recognition of our dataset's value for indoor geolocation tasks. Below, we address each of the raised concerns point by point.
>
>  1. ***This is a dataset paper, but the methodological part has been widely overlooked. When introducing a new dataset, one should expect extensive benchmarking of methods from the literature (Regression, Classification, hybrid [2], generative [3], retrieval [5], etc.). Finetuning Geoclip is not enough.***
>
> Thank you for raising the need for broader methodological benchmarking. In Section 5 of
> the revised manuscript (lines 334), we expand our evaluation to span other paradigms. These include:
> - Retrieval-based methods (CLIP, DINOv2, ResNet, StreetCLIP,  IndoorCLIP) with results in Table 2 (line 365).
>  - Hybrid/alignment-based models (GeoCLIP∗, PIGEON∗, Translocator∗) in Table 3(line 378).
>  - Generative/VLM-style models (LLaVA-1.6, InternVL2-8B) in Table 4 (lines 388).
>  - Cross-dataset generalization of GeoCLIP* to im2gps3k, yfcc4k, and yfcc26k appears in Table 5 (line 445), to demonstrate transferability beyond our indoor test set.
>
>
> 2. ***Recently, LLMs have been showing strong results for geolocation, and I would like to see extensive benchmarking of the different LLMs (at least the open-source ones).***
>
> Thank you for pointing this out. Kindly refer to Table 4 (line 388) in our revised manuscript for benchmarking results of two generative models  (LLaVA-1.6 and InternVL2-8B).
>
>
>  3. ***The test set is temporally separated, but it should be spatially separated (as in [2]). If having the same room in train and test is important, there must be a mechanism to ensure the pictures are sufficiently different. (It's not because a picture is uploaded later that it's not the same image).***
>
> Thank you for stressing the importance of spatial (not only temporal) separation. In the revised manuscript, we clarify that our pre-processing pipeline already enforces strict spatial and visual decontamination before separating the test set. First, we apply pHash-based duplicate removal to eliminate near-identical or trivially modified images, Then, we remove room-level redundancy by grouping images that share latitude/longitude, date, and scene category, and retaining only a single representative per group (lines 185-189). Importantly, this entire procedure is performed before dataset splitting, ensuring that no room-level or visually overlapping scenes propagate into the benchmark test set.
>
>
> 4. ***Similarly, it would be interesting to benchmark the different visual encoders. Are there some that are more accurate on indoor scenes than outdoor?***
>
> Thank you for this suggestion. In the revised manuscript, we benchmark a range of widely used visual encoders in a zero-shot retrieval setting to assess whether some backbones perform better on indoor imagery. As detailed in Section 5 (lines 270–289), we evaluate CLIP ViT-B/32, CLIP ViT-
> L/14, DINOv2-L, ResNet-50, StreetCLIP, using nearest neighbour retrieval , all without fine-tuning to isolate encoder effects. The comparative results are reported in Table 2 (line 365) with StreetCLIP outperforming other frozen encoders.
>
>
> 5. ***Missing Citations***
>
> We appreciate the reviewer’s careful attention to recent geolocation work. In the revised manuscript, we have included citations to GOMAA-Geo, OpenStreetView-5M, Around the World in 80 Timesteps, GaGA, and G3, and have expanded the related-work discussion to reflect their relevance. In particular, we draw a lot of inspiration from the evaluation methodology introduced in OpenStreetView-5M, as suggested.
>
>
> 6. ***I think the paper aesthetics could be improved. In particular, the tables are pretty unaesthetic***
>
> Thank you for the feedback regarding table aesthetics. In the revised manuscript, we have updated all tables for improved readability and aesthetics.
>
> We hope you will kindly consider the revisions we have made to our manuscript as recommended in your review.

---

### Author Response · Authors · 2025-11-27
**Summary of Revisions**

We would like to sincerely thank all reviewers for the time and attention they devoted to evaluating our submission. We appreciate their constructive feedback and are grateful for the acknowledgement that a large-scale indoor geolocation dataset can be valuable not only for advancing research, but also for real-world applications such as human-tracking investigations. Your comments have helped us strengthen the clarity, presentation and the technical depth of the work.

In response to the reviews, we have made some updates to the paper:

* Clarified how INDOOR-3.6M is constructed and how it relates to the INDOOR-40K benchmark split.
* Expanded and refined the methodological evaluation, with more baselines, including retrieval, hybrid geolocation methods and VLMs.
* Improved table and figure descriptions and including more error-distribution analysis.
* Improved the readability and overall presentation of the manuscript.

We will address each reviewer's comments individually in more detail.

---

### Author Response · Authors · 2025-12-03
**Revision Summary**

We sincerely thank all reviewers for their constructive feedback, which has significantly strengthened our paper. We appreciate the recognition of the dataset’s value for advancing indoor geolocation research, including its relevance for real-world investigative and forensic applications.

**INDOOR-3.6M addresses a clear and underexplored problem.**
Existing geolocation benchmarks focus almost entirely on outdoor imagery, whereas indoor visual geolocation remains comparatively understudied. This work presents the first systematic, global-scale study of indoor geolocation along with systematic evaluation and benchmarking for this important setting.

We provide:

* **The first large-scale indoor geolocation dataset:**
  3.6 Million diverse indoor images across 220+ countries/territories with rich metadata.

* **A geographically representative benchmark:**
  INDOOR-40K, constructed using a principled sampling framework balancing population, land area, and visual diversity.

* **A comprehensive evaluation suite:**
  The first systematic assessment across retrieval, hybrid, and VLM paradigms, demonstrating that coarse indoor geolocation (e.g., 66% continent accuracy) is feasible, while finer-grained localization remains a challenging open problem.

## **Improvements Made in Response to Reviewers**

These revisions strengthen clarity and reproducibility without altering the core contribution:

* **Expanded baselines (Tables 2–4):**
  Added retrieval models (CLIP, DINOv2, StreetCLIP, IndoorCLIP), hybrid methods (PIGEON*, TransLocator*), and VLM baselines (LLaVA-1.6, InternVL2-8B), plus cross-dataset generalization results (Table 5). These contextualize model performance without changing our original conclusions.

* **Improved clarity and presentation:**
  Clarified dataset construction, spatial de-duplication, sampling methodology, and enhanced tables and figures.

* **Reproducibility enhancements:**
  Appendix A.2 now includes hyperparameters, data sources, and training details.

* **Requested ablation:**
  Added the full P(indoor) ablation (0.5–0.9), demonstrating stable performance and addressing concerns about ambiguous indoor-likelihood ranges.


## Dataset Dissemination

Our dissemination approach follows established precedent in the community.

* **Flickr and Wikidata images** will be hosted directly, with URLs shared as they are provided under creative commons licenses.
* **Hotel website images** are provided as URLs with accompanying download scripts. This approach mirrors datasets such as Hotels-50K (Stylianou et al.,2019), which follow the same methodology for non-commercial academic research. To ensure reproducibility and prevent the issue of broken links, these URLs will also be archived via the Internet Archive.

We believe the paper provides a valuable foundation for future research in this underexplored domain and we appreciate your consideration.

---

### Meta-Review · Area_Chair_Jkhb · 2026-01-02

**Summary:**

Concerns center on whether the work meets the ICLR bar as a dataset paper. Major issues include weak and narrow benchmarking (largely centered on GeoCLIP), significant clarity and presentation problems, confusion around dataset splits and domain mismatch between training and evaluation, and limited validation of the claimed metadata value. Several reviewers also questioned dataset curation details, reproducibility, and release clarity. Overall, while promising, the paper does not yet feel mature or rigorous enough for acceptance.

**Reviewer Concerns:**

No rebuttal provided

**Reviewer Scores:**

No rebuttal provided

---

### Decision · Program_Chairs · 2026-01-26

Reject